# ECR-MobileNet: An Imbalanced Largemouth Bass Parameter Prediction Model with Adaptive Contrastive Regression and Dependency-Graph Pruning

**DOI:** 10.3390/ani15162443

**Published:** 2025-08-20

**Authors:** Hao Peng, Cheng Ouyang, Lin Yang, Jingtao Deng, Mingyu Tan, Yahui Luo, Wenwu Hu, Pin Jiang, Yi Wang

**Affiliations:** 1College of Information and Intelligence, Hunan Agricultural University, Changsha 410128, China; sx20230201@stu.hunau.edu.cn (H.P.); ouyang@stu.hunau.edu.cn (C.O.); 2767959117@stu.hunau.edu.cn (L.Y.); 984711035@stu.hunau.edu.cn (J.D.); 19907426171@stu.hunau.edu.cn (M.T.); 2College of Mechanical and Electrical Engineering, Hunan Agricultural University, Changsha 410128, China; luoyh@hunau.edu.cn (Y.L.); 1233087@hunau.edu.cn (W.H.); 1233032@hunau.edu.cn (P.J.)

**Keywords:** body length and weight prediction, imbalanced regression, lightweight neural networks, deep learning, smart aquaculture

## Abstract

Accurately tracking the growth of fish is essential for profitable and sustainable fish farming. However, the traditional method of catching and measuring fish by hand is slow, costly, and highly stressful for the animals, which can harm their health and slow their growth. To solve this, we developed a new artificial intelligence (AI) system called ECR-MobileNet. It acts like a “smart camera,” instantly and accurately estimating the length and weight of a largemouth bass from just a picture. Our AI is uniquely designed to be both precise and extremely efficient. It intelligently focuses on the most important parts of the fish’s image and is specially trained to be fair and accurate for fish of all sizes, from the smallest to the largest. We then streamlined this AI to make it so lightweight that it can run on small, affordable computers. Our method proved to be more accurate than 14 other modern AI models, providing near-instant results. This technology offers a stress-free way to monitor fish, helping farmers use feed more efficiently, reduce waste, and improve their profits and making modern, intelligent aquaculture a practical and accessible reality for farms of all sizes.

## 1. Introduction

In recent years, the aquaculture industry has undergone rapid global expansion, particularly in the culture of high-value fish species [1]. The industrial scale of largemouth bass (*Micropterus salmoides*), a species of immense economic value in global aquaculture, continues to expand worldwide. According to the China Fishery Statistical Yearbook (2024), the production of farmed largemouth bass in China reached 852,000 t in 2023, accounting for the vast majority of global output [2]. Within intensive aquaculture systems, the ability to precisely monitor the growth status of bass has become a central requirement for driving the industry’s intelligent transformation. This is especially true for the non-invasive, real-time prediction of key biological indicators such as body length and weight. Accurate growth data provide a direct basis for formulating scientific feeding strategies, effectively reducing feed costs, and improving the feed conversion ratio (FCR) [3]; furthermore, such data are fundamental for yield estimation and market planning. These factors directly influence the profitability, resource utilization efficiency, and market competitiveness of aquaculture enterprises [4]. Therefore, developing efficient and accurate technologies for growth prediction is a critical step in advancing the transition of the largemouth bass industry toward intelligent, precision-oriented, and sustainable practices [5].

Presently, the conventional measurement of bass body length and weight remains heavily reliant on manual sampling, a procedure that typically involves netting, anesthetizing, and physically handling the fish. This approach presents significant limitations. First, it is exceedingly labor-intensive and inefficient. For instance, sampling a 3.3-hectare (approximately 50-mu) pond requires 3–5 workers for over four hours, incurring high labor costs and rendering a full-pond census in large-scale farms practically infeasible [6]. Second, netting and direct human contact induce severe stress responses in fish, which have led to annual yield losses of 5–10% and increased susceptibility to pathogen infections [7]. These stressors can also significantly suppress feeding behavior and growth rates, and even elevate mortality, thereby directly compromising production performance [8]. Finally, the reliability of manual measurements is often compromised by their dependence on operator experience, making it difficult to ensure data repeatability and accuracy [9]. While early research attempted to address these issues using traditional mathematical models based on morphological parameters, these models were constrained by their reliance on limited, manually collected data samples. Consequently, they struggled to capture the complex non-linear relationship between body length and weight, exhibited poor generalization ability, and ultimately failed to eliminate the dependence on conventional measurement methods [10].

The rapid development of computer vision (CV) and deep learning technologies in recent years has offered non-contact solutions to address the aforementioned challenges [11]. Early CV techniques, which relied on traditional image processing to extract contour features, demonstrated poor robustness in underwater environments characterized by complex backgrounds and fluctuating lighting conditions [12]. With the advent of deep learning, convolutional neural networks (CNNs) have spurred significant progress in this domain, owing to their end-to-end feature learning capabilities [13]. From the classic AlexNet [14] to deeper architectures like ResNet [15], CNN models have consistently demonstrated superior performance across a range of agricultural image recognition tasks [16]. More recently, the vision transformer (ViT) has been introduced for biometric prediction, leveraging its capacity for global feature capture [17]. However, its high computational cost restricts its practical application on resource-constrained edge devices [18].

To address this challenge, model lightweighting has emerged as a key research focus, aiming to reduce parameter counts and computational complexity while maintaining accuracy [19]. Its application is particularly crucial in the context of aquaculture. This has led to the development of lightweight networks such as MobileNet [20,21,22], ShuffleNet [23,24], and GhostNet [25,26,27]. Zhang et al. [28] combined pose recognition with binocular vision to predict fish weight, achieving an error rate as low as 2.87%; however, their model was not optimized for edge deployment. Jansi Rani et al. [29] utilized YOLOv8 and preprocessing techniques to tackle the problem of turbid water, but the model modifications increased the computational load. Yan et al. [30] developed a lightweight YOLOv5m to meet the demands of underwater robots, yet their work did not focus on the prediction of growth indicators. Zheng et al. [31] employed video object segmentation to improve recognition rates in complex environments, albeit at a high computational cost. Xue et al. [32] used 3D imaging to predict the swimming speed of rainbow trout, but the complex workflow rendered the method unsuitable for real-time applications.

However, despite these explorations into diverse model architectures and application scenarios, existing studies have generally overlooked two critical issues. The first is the prevalent problem of sample size imbalance in aquaculture datasets, where samples of medium-sized fish far outnumber those of extreme sizes. This imbalance can impair the model’s predictive performance on long-tail data. Second, how to systematically achieve extreme model lightweighting to meet the demands of future large-scale deployment on low-cost edge devices remains an open challenge.

To address the aforementioned challenges, this study proposes a novel deep learning framework named ECR-MobileNet, designed for the high-precision and high-efficiency prediction of the body length and weight of largemouth bass. The main contributions of this paper are as follows:We designed and introduced the adaptive multi-scale contrastive regression (AMCR) loss function, specifically tailored for imbalanced regression, into the task of fish biometric parameter prediction. Combined with the efficient channel attention (ECA) module, this approach significantly improved the model’s predictive accuracy on long-tail data and its overall robustness.We proposed and validated a complete technical pipeline, from high-accuracy model training to extreme lightweight deployment. By employing a general structured pruning technique based on dependency graphs (DepGraph), implemented via the Torch-Pruning (v1.5.2) library, we substantially compressed the model’s parameter count and computational load by 44.1% and 41.7%, respectively, while maintaining or even improving prediction accuracy.Through extensive comparative experiments and ablation studies, we systematically validated the independent contributions and synergistic effects of each innovative module within our proposed framework. This work provides solid empirical evidence and establishes a new performance benchmark for future model design and optimization strategies in this field. The comprehensive performance of our method on the bass biometric prediction task is illustrated in Figure 1, visually substantiating the above contributions.

## 2. Materials and Methods

### 2.1. Dataset Description

The accurate acquisition of fish biometric parameters is a critical prerequisite for overcoming key technological bottlenecks in intelligent aquaculture. To address the limitations of conventional manual measurements—namely, their inefficiency and stress-inducing nature—and the poor generalization of existing computer vision models in authentic aquaculture environments (e.g., fluctuating illumination, water turbidity, and complex fish poses). Systematically covering the entire value chain from farming to market distribution, this dataset is designed to provide a foundational resource for the development of highly robust models for growth parameter prediction.

#### 2.1.1. Data Acquisition and Scenario Design

A total of 120 healthy largemouth bass samples (ranging from 158 to 920 g) were systematically collected from aquaculture environments in Xiangtan, Changsha, and Jishou, Hunan Province. These environments included high-density earthen ponds (in Xiangtan) and indoor recirculating concrete tanks (in Jishou), with water temperatures maintained between 22 and 28 °C and pH levels of 7.0–8.5. Based on the characteristics of key stages in the aquaculture value chain, three distinct scientific scenarios were designed:Standardized lateral view for on-farm monitoring: In this scenario, we placed the fish in a glass tank and captured images laterally from the outside of the tank. This approach was designed to acquire high-quality images with a clean background, uniform lighting, and relatively standard fish poses, serving as ideal conditions for the model to learn baseline morphological data. This scenario was conducted in a controlled environment at the Xiangtan aquaculture base with the objective of acquiring standard lateral-view images rich in information for precise modeling. We captured images of 60 fish using a ZED 2i stereo camera (Stereolabs, San Francisco, CA, USA). To challenge and enhance the model’s environmental adaptability, this scenario specifically included two critical conditions: clear water and simulated turbid water (see Figure 2a). This was intended to train the model to cope with varying levels of water transparency. In this stage, approximately 660 high-quality data entries, each comprising an RGB image and its corresponding depth map, were collected.Simulated retail environment for market-chain tracking: In contrast to Scenario 1, this scenario utilized a waterproof high-definition camera to capture footage underwater, inside a tank that simulated a retail environment. This design was intended to replicate more challenging real-world conditions, including light refraction from underwater imaging, dynamic illumination from surface reflections, more complex backgrounds, and the uncooperative, non-rigid poses of fish in a confined space. We procured 30 live specimens from an aquatic market in Changsha and recorded underwater lateral-view videos using a waterproof high-definition camera (Fuli, Model A8, Shenzhen, China). This scenario simulated the temporary holding tanks common at retail terminals. The core challenges were complex dynamic lighting (e.g., surface reflections), diverse backgrounds (retail tanks), and uncooperative, non-rigid poses adopted by the fish due to spatial constraints. Using key-frame extraction techniques, we obtained approximately 340 dynamic images featuring diverse poses (see Figure 2b), which significantly enhanced the dataset’s diversity and robustness for complex real-world scenarios.Top-down morphological view for on-farm biomass estimation: At the Jishou aquaculture base, we again utilized the ZED 2i camera to capture images of 30 fish from a direct top-down perspective (see Figure 2c). This viewpoint is crucial for biomass estimation and is intended to complement the information provided by lateral views. The fish width and dorsal contour data available from this perspective are key variables for accurately estimating the condition factor and body weight. This provided the necessary data dimensionality for the model to overcome the limitations of a single viewpoint and establish a more precise non-linear mapping between body length and weight.

#### 2.1.2. Ground Truth Annotation and Quality Control

Immediately following image acquisition, physical measurements were performed on each fish to acquire its ground truth (GT) data. To ensure accuracy and minimize stress, fish were briefly sedated in a buffered tricaine methanesulfonate (MS-222) solution at a concentration of approximately 100 mg/L. The exposure time to the sedative was around 2–3 min. The handling time out of water was kept under 60 s per fish. Body weight was recorded to the nearest gram using an electronic scale, and total length—the straight-line distance from the snout tip to the end of the caudal fin—was measured to the nearest millimeter using a standard measuring tape. Following measurements, fish were transferred to a dedicated recovery tank with fresh, aerated water and monitored until normal activity resumed, typically within 5–10 min, before being returned to their holding tanks.

All measurement data were rigorously linked to their corresponding images via a unique identifier. Specifically, each fish was assigned a numbered tag, which was recorded alongside its measurements and was visible in the initial photograph of each sequence to ensure a foolproof one-to-one correspondence. To ensure data quality, we employed a two-stage quality assurance (QA) protocol. The first stage involved a rapid screening process to discard images exhibiting severe motion blur, partial occlusion greater than 30%, or significant overexposure. The second stage consisted of a meticulous review of the remaining images to confirm that fish contours were clear and features were intact, leading to the removal of any ambiguous or compromised samples. Ultimately, this QA protocol resulted in a final dataset of 1284 high-quality images retained from an initial pool of 1354.

#### 2.1.3. Dataset Statistical Analysis and Partitioning

The final dataset exhibited broad coverage in terms of sample size. The total length ranged from 17.5 to 34.0 cm (mean: 24.64 cm, standard deviation [SD]: 2.58 cm), while the body weight ranged from 158 to 920 g (mean: 420 g, SD: 120 g). Notably, the coefficient of variation (CV) for weight (28.6%) was significantly higher than that for total length (10.5%). This disparity indicates substantial individual variation in body condition, even among fish of similar lengths. This wide distribution is by design, as the samples were intentionally selected to span the full growth cycle from advanced fingerlings to market-ready adults, thereby enabling the development of a model capable of continuous growth monitoring. This characteristic not only provides a rich data foundation for the model to learn the complex, non-linear relationship between length and weight but also constitutes a core challenge of this research.

To ensure an unbiased model evaluation, we employed a stratified sampling strategy based on total length to partition the dataset into training, validation, and test sets at a ratio of 8:1:1. This approach ensured that the distribution of key biometric indicators in each subset was consistent with that of the overall dataset. The detailed partitioning and statistics are presented in Table 1.

#### 2.1.4. Data Preparation and Augmentation

To ensure the uniformity of input data, enhance model generalization, and mitigate the risk of overfitting on a limited dataset, we constructed a systematic data preparation and augmentation pipeline. This pipeline, strictly adhering to standard principles of machine learning data processing, sequentially performed data cleaning and alignment, targeted data augmentation, and normalization. Crucially, the training set was treated differently from the validation and test sets to ensure an objective and unbiased evaluation of the model’s performance.

For the training set, we employed a composite strategy that combined geometric and photometric transformations to simulate the natural morphological variations and lighting disturbances found in authentic aquaculture environments. On the geometric level, random horizontal flipping was applied to enhance the model’s invariance to fish orientation, while random rotation simulated the natural tilting poses of fish in water. At the photometric level, color jittering was used to simulate color deviations under varying water quality and lighting conditions. Furthermore, random grayscale conversion enhanced the model’s robustness in scenarios with color distortion by reducing its over-reliance on color information.

In addition, to support the contrastive regularization training strategy, we further introduced asymmetric data augmentation. This process involved generating a pair of views for each training image with differing levels of perturbation. The two views remained semantically consistent but possessed subtle differences in their visual representations, thereby constructing positive pairs to strengthen the model’s ability to learn intrinsic feature associations.

Upon conversion to tensors, all images were normalized using the mean [0.485, 0.456, 0.406] and standard deviation [0.229, 0.224, 0.225] derived from the ImageNet dataset. This process stabilizes the model training and accelerates convergence. To address the significant disparities in physical units and numerical ranges between the two regression targets (total length and weight), this study performed label standardization. The standardization parameters (i.e., mean and standard deviation) were fitted exclusively on the training set labels and subsequently applied across all data subsets. This strategy ensures a balanced contribution of different physical scales to the loss function during multi-task regression.

Table 2 summarizes the specific pipeline and parameter settings for data augmentation and preprocessing. The training set underwent a series of augmentations—including geometric flipping, rotation, color jittering, and grayscaling—to expand its diversity. In contrast, the validation and test sets were subjected only to consistent resizing and normalization to avoid introducing additional perturbations.

In summary, rigorous data cleaning and alignment ensured the integrity and consistency between the input images and their corresponding output labels. By employing a differentiated augmentation strategy and proper standardization, we effectively expanded data diversity while strictly controlling the risk of data leakage. This provides a high-quality, standardized data foundation for the subsequent training and evaluation of the high-precision, lightweight models for fish growth parameter prediction.

### 2.2. AMCR: Adaptive Multi-Scale Contrastive Regression Loss Function

Inspired by the InfoNCE [33] and ConR [34] methods, the AMCR loss function is specifically designed and optimized for the characteristics of deep regression tasks, effectively enhancing the accuracy and robustness of biometric parameter prediction.

In the context of predicting the length and weight of largemouth bass, AMCR first generates semantically consistent positive pairs by applying a series of augmentations—such as cropping, rotation, and color jittering—to the original fish images. It then dynamically constructs anchor-positive–negative relationships based on the similarity of the ground truth labels. By fusing supervised contrastive learning with information from the label space, the AMCR loss function drives samples with similar lengths and weights to cluster together in the feature space while pushing samples with significant biometric differences apart. This mechanism enhances the model’s generalization ability, particularly for imbalanced data distributions.

#### 2.2.1. Problem Formulation

This research addresses the problem of non-contact, high-precision, computer vision-based prediction of key biometric parameters (i.e., total length and body weight) for largemouth bass in aquaculture. Let $D={\left(x_{i},y_{i}\right)}_{i=1}^{N}$ be a training dataset of N samples, where $x_{i}$ represents the i−th input image and $y_{i}$ is its corresponding ground truth label vector. The label vector $y_{i}$ comprises two key biometric indicators: $y_{i}={\left[l_{i},w_{i}\right]}^{T}$, where $l_{i}$ denotes the total length and $w_{i}$ represents the body weight. Consequently, the label space is two-dimensional continuous space, making this a multi-dimensional regression task. This study places a special focus on a prevalent challenge in authentic aquaculture environments: the imbalanced distribution of labels. Specifically, the number of samples representing medium-sized bass in the dataset far outnumbers those of extreme sizes (i.e., very small or very large), causing the distribution of labels $y_{i}$ to exhibit a long-tailed characteristic. We define a deep learning model as $M\left(\cdot \right)=R\left(E\left(\cdot \right)\right)$, which is composed of a feature extractor, $E\left(\cdot \right)$, and a regressor, $R\left(\cdot \right)$. The objective of this research is to train the model M such that for any given input image $x_{i}$, its predicted output, ${\hat{y}}_{i}=M\left(x_{i}\right)$, closely aligns with the corresponding ground truth label, $y_{i}$.

#### 2.2.2. Imbalanced Contrastive Regression

In regression tasks, the structure of the label space is expected to reflect semantic relationships within the feature space. However, an imbalanced data distribution disrupts this mapping, leading to a confounding effect where the feature statistics of minority and majority samples overlap, even when their ground truth labels differ significantly [35]. To address this challenge, we propose the AMCR framework. It innovatively integrates supervised information from the label space into a contrastive learning paradigm, with the goal of constructing a distributionally balanced feature space and thereby establishing a more robust feature similarity metric.

In essence, AMCR is a continuous extension of the InfoNCE loss. First, to introduce sample diversity, we apply data augmentation to each input image $x_{j}$ to generate two augmented views. The resulting set of augmented samples is defined as ${\left(x_{j}^{a},y_{j}\right)}_{j=1}^{2N}$, where $x_{j}^{a}$ is an augmented image, and $y_{j}$ is its corresponding original ground truth label. The core of AMCR lies in its dynamic strategy for selecting positive and negative pairs. For each augmented sample $\left(x_{j}^{a},y_{j}\right)$, AMCR first determines its positive and negative counterparts. Since the label $y_{j}$ remains unchanged after augmentation, every sample has at least one positive pair (i.e., the other view from the same original image). If a sample also has at least one negative pair, it is treated as an ‘anchor’. The regularization process of AMCR then pulls this anchor closer to its positive pairs in the feature space while simultaneously pushing it away from its negative pairs. This core concept is illustrated in Figure 3.

Pair Selection. Given a pair of augmented samples, $\left(x_{i}^{a},y_{i}\right)$ and $\left(x_{j}^{a},y_{j}\right)$, each sample is first passed through the feature encoder E(·) to extract feature vectors $z_{i}$ and $z_{j}$. These are subsequently fed into the regressor R(·) to obtain the predicted values ${\hat{y}}_{i}$ and ${\hat{y}}_{j}$. The predicted values $\left({\hat{y}}_{i},\;{\hat{y}}_{j}\right)$ and the ground truth labels $\left(y_{i},{\;y}_{j}\right)$ are jointly used to determine whether the two samples should form a positive pair, a negative pair, or not be paired at all. The detailed decision-making process is illustrated in Figure 4. To measure the similarity between multi-dimensional labels, we made a key extension to the basic contrastive regression method. We first define a similarity threshold, ω. Given a multi-dimensional distance function, Dist(·,·), if the distance between two label vectors, $y_{i}$ and $y_{j}$, satisfies Dist $\mathrm{Dist}(y_{i},y_{j})\leq \omega$, they are considered similar, denoted as $y_{i}\approx y_{j}$. In this study, we employed the Manhattan distance to compute the combined difference across the total length (l) and body weight (w) dimensions: $\mathrm{Dist}(y_{i},y_{j})=|l_{i}-l_{j}|+\left|w_{i}-w_{j}\right|$. As shown in Figure 4, based on the similarity measure defined above, the pairing rules are defined as follows:

Positive pair: formed if and only if the ground truth labels of the two samples are similar, i.e., $y_{i}\approx y_{j}$.

Negative pair: formed when the ground truth labels of the two samples are dissimilar $\mathrm{Dist}(y_{i},y_{j})>\omega$ but their model predictions are erroneously judged to be similar $\mathrm{Dist}({\hat{y}}_{i},{\hat{y}}_{j})\leq \omega )$.

No pair: when neither of the above conditions is met, the samples do not participate in contrastive learning.

Anchor selection. For each augmented sample $\left(x_{j}^{a},y_{j}\right)$, the set of all its positive feature vectors is denoted as ${K^{+}}_{j}={z_{p}}^{{N^{+}}_{j}}$_j_, and the set of all its negative feature vectors is denoted as ${K^{-}}_{j}={z_{q}}^{{N^{-}}_{j}}$, where ${N^{+}}_{j}$ and ${N^{-}}_{j}$ represent the number of positive and negative samples, respectively. If ${N^{-}}_{j}>0$, the sample $\left(x_{j}^{a},y_{j}\right)$ is selected as an anchor and participates in the AMCR regularization process.

**Figure 4 animals-15-02443-f004:**
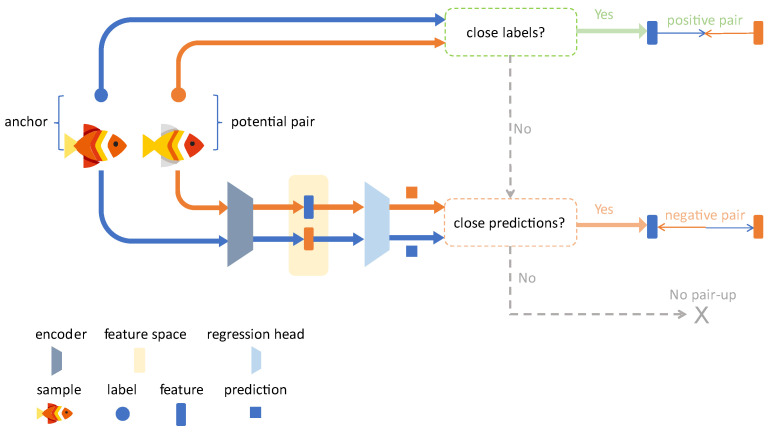
Workflow of the pair selection mechanism in AMCR. (1) Positive pair: a positive pair is formed if the ground truth labels y are similar $Dist\left(y_{i},y_{j}\right)\leq \omega$. (2) Negative pair: a negative pair is assigned if the positive pair condition is not met, but the predicted values are erroneously judged to be similar $Dist\left({\hat{y}}_{i},{\hat{y}}_{j}\right)\leq \omega$, compelling the model to correct this error. (3) No pair: in all other cases, the pair does not participate in the contrastive loss computation.

For each candidate anchor sample j, AMCR defines its contrastive loss as $L_{{\mathrm{AMCR}}_{j}}$. If sample j is not selected as an anchor, then $L_{{\mathrm{AMCR}}_{j}}=0$. If sample j is selected as an anchor $\left({N^{-}}_{j}>0\right)$ $L_{{\mathrm{AMCR}}_{j}}$ concurrently performs two optimization tasks: it pulls the anchor $z_{j}$ closer to its positive samples $z_{p}\in {K^{+}}_{j}$, and it dynamically repels the negative samples based on their label dissimilarity. Specifically, the repulsive force exerted on a negative sample $z_{q}\in {K^{-}}_{j}$ is proportional to $\mathrm{Dist}\left(y_{j},y_{q}\right)$, meaning that negative samples with greater label dissimilarity are pushed away more forcefully.(1)$$L_{\mathrm{AMCRj}}=-\log\frac{1}{N_{j}^{+}}\underset{z_{i}\in K_{j}^{+}}{\sum }\frac{\exp(z_{j}\cdot z_{i}/\tau )}{\sum_{z_{p}\in K_{j}^{+}}\exp(z_{j}\cdot z_{p}/\tau )+{\sum_{z_{q}\in K_{j}^{-}}S}_{j,q}\exp(z_{j}\cdot z_{q}/\tau )}$$
where τ is a temperature hyperparameter. The term $\alpha_{j,q}$,$\omega_{j,q}$ is a repulsion weight for each negative pair, defined as:(2)$$S_{j,q}=f_{S}\left(\eta_{j},\mathrm{Sim}\left(y_{j},y_{q}\right)\right)$$
where $y_{q}$ is the ground truth label corresponding to the feature vector $z_{q}=E\left(x_{q}^{a}\right)$. The repulsion strength $\eta_{j}$ for each sample $\left(x_{j}^{a},y_{j}\right)$ is dynamically adjusted by the label distribution $D_{y}$ to enhance optimization for minority samples; here, $\eta_{j}\propto d_{j}$, where $d_{j}$ is a density weight calculated from the empirical label distribution. The similarity function $f_{S}$ generates the coefficient $S_{j,q}$, which satisfies a dual constraint: $S_{j,q}\propto \eta_{j}$ and $S_{j,q}\propto 1/\mathrm{Sim}\left(y_{j},y_{q}\right)$. Finally, the overall AMCR regularization term is the average of the anchor losses over the entire set of augmented samples:(3)$$L_{\mathrm{AMCR}}=\frac{1}{2N}\overset{2N}{\underset{j=0}{\sum }}L_{\mathrm{AMCRj}}$$

#### 2.2.3. Theoretical Insight

We followed the theoretical analysis framework from the original contrastive learning literature to theoretically validate the effectiveness of our proposed AMCR method. We derived a theoretical upper bound for the probability of mislabeling on minority class samples, which is constrained by our proposed $L_{\mathrm{AMCR}}$ loss function.(4)$$\frac{1}{4N^{2}}\overset{2N}{\underset{j=0,x_{j}\in A}{\sum }}\;\overset{K_{j}^{-}}{\underset{q=0}{\sum }}\log S_{j,q}p({\hat{Y}}_{j}|x_{q})\leq L_{\mathrm{AMCR}}+\epsilon ,\;\epsilon \;\overset{N\to \infty }{\to }\;0$$

Here, A is the set of anchors, and xq is a negative sample image. The term p(Y^j|xq) represents the probability that the predicted value Y^j of a negative sample xq erroneously falls within the multi-dimensional similarity region Y^j centered at the anchor’s predicted value Y^j with radius ω. This similarity region is defined as Y^j=y∈ℝ2|Disty,y^j≤ω. We interpret p(Y^j|xq) as the probability of “prediction collapse” for sample xq. The left-hand side of inequality (4) represents the weighted sum of these prediction collapse probabilities for all negative pairs during training. This formulation explicitly reveals that the AMCR loss, LAMCR, places an upper bound on this total probability. Therefore, minimizing LAMCR necessarily leads to a reduction in both the number of anchors (i.e., misclassified pairs) and the degree of prediction collapse.

Furthermore, each collapse probability $p({\hat{Y}}_{j}|x_{q})$ is modulated by the weight $S_{j,q}$, whose value is proportional to the ground truth label dissimilarity $\mathrm{Dist}\left(y_{j},y_{q}\right)$. This mechanism implements a differentiated penalty for severe prediction errors. Consequently, optimizing our proposed AMCR loss can significantly reduce the misclassification probability for minority samples. By strengthening the representation learning for minority classes through this weighting mechanism, it enhances the model’s overall performance and predictive fairness.

### 2.3. Overall Model Architecture

To achieve non-contact prediction of the body length and weight of largemouth bass, this study proposes a lightweight deep learning model named ECR-MobileNet (Efficient Channel Attention Regression MobileNet). The model adheres to the standard backbone-and-head design paradigm. It utilizes MobileNetV3-Small as its core backbone network, integrated with an ECA module and a simple linear regression head. This architecture strikes a balance between high-precision prediction and low computational complexity, making it particularly suitable for real-time fish growth monitoring on resource-constrained edge devices. As illustrated in Figure 5, the architecture primarily consists of two components: the backbone network and the regression head.

#### 2.3.1. Feature Extraction Backbone

The choice of a backbone network is a core determinant of a model’s performance and efficiency. In aquaculture applications, models must satisfy the dual requirements of high inference speed and low computational resource consumption while ensuring robust feature extraction capabilities. To this end, our study selected MobileNetV3-Small, a benchmark model for lightweight networks, as the primary feature extraction backbone. This choice not only represents an excellent balance between efficiency and performance but is also highly suitable for real-time applications in resource-constrained environments.

The central advantage of MobileNetV3 lies in its relentless pursuit of computational efficiency, achieved through several key design innovations:

Depth-wise separable convolutions: This technique decomposes a standard convolution into two distinct steps: a depth-wise convolution and a point-wise convolution. This approach achieves a feature extraction efficacy comparable to that of standard convolutions while significantly reducing both the parameter count and the computational load (FLOPs).

Inverted residuals and linear bottlenecks: This structure employs an “expand-convolve-compress” feature transformation pattern, where feature extraction and processing occur in a high-dimensional space. Simultaneously, linear bottleneck layers and residual connections are introduced at the low-dimensional input and output stages. This design effectively preserves the integrity of the information flow and mitigates the potential information loss caused by the ReLU activation function in low-dimensional spaces, thereby enhancing the quality of feature representations.

NAS-based architecture search: The architecture of MobileNetV3 is not entirely handcrafted; instead, it was optimized using neural architecture search (NAS) techniques tailored to the latency characteristics of specific hardware, such as mobile CPUs. This ensures that the model achieves a Pareto-optimal trade-off between speed and accuracy on real-world devices, providing strong support for practical deployment.

Departing from the common practice of relying on large pre-trained models, this study adopted a strategy of training the model from scratch, without using pre-trained weights from general-purpose datasets like ImageNet. Although this choice poses greater challenges for model convergence, its advantage is that the features extracted during training are exclusively tailored to our proprietary largemouth bass dataset. This avoids potential performance bottlenecks that can arise from domain discrepancies in pre-trained models. Training from scratch provides a more rigorous experimental foundation for validating the effectiveness of our proposed overall framework on task-specific data.

#### 2.3.2. Efficient Channel Attention Regression Head

Conventional regression heads typically employ one or more fully connected (FC) layers to directly map a feature vector to the output values. This design treats each channel in the feature vector equally, failing to account for the significant differences in information content across various channels. In fine-grained tasks such as bass parameter prediction, certain feature channels often contain more discriminative information and contribute more significantly to the final prediction than others.

To enable the model to adaptively focus on the feature channels most critical to the current prediction task, this study introduces a key enhancement to the regression head design by incorporating the ECA mechanism. Proposed in ECA-Net [36], the core idea of the ECA mechanism is to significantly optimize the computational efficiency of the attention module while maintaining its performance. Compared to the Squeeze-and-Excitation (SE) attention mechanism, which is widely used in models like MobileNetV3, ECA offers two notable advantages:Avoidance of dimensionality reduction bottlenecks: The SE module facilitates channel interaction through two FC layers: the first compresses the channel dimension, and the second restores it. However, the authors of ECA-Net demonstrated through experiments that this dimensionality reduction is not essential for learning effective channel attention and may even be detrimental to model performance. Consequently, the ECA module discards the dimensionality reduction step and performs interactions directly on the original channel dimensions, thereby eliminating this potential bottleneck.Efficient local cross-channel interaction: ECA proposes an efficient local interaction strategy where the attention weight for each channel is learned by considering information from only its k nearest neighboring channels, rather than engaging in global interaction with all channels. This strategy can be implemented with high efficiency using a one-dimensional (1D) convolution with a kernel size of k

Furthermore, the authors of ECA-Net proposed that this interaction range, k, should not be fixed but should instead be proportional to the channel dimension C. Thus, the kernel size k is adaptively determined via the following mapping:(5)$$k={\left|\frac{\log_{2}C+b}{\gamma }\right|}_{\mathrm{odd}}$$
where $|\;\cdot \;{|}_{\mathrm{odd}}$ denotes the nearest odd number, and γ and b are tunable hyperparameters. This adaptive strategy enables the model to automatically select the optimal local interaction range for different network layers, thereby further enhancing model performance and generalization ability.

In our study, the final regression head was designed as follows: The feature vector $z_{i}$ output by the backbone network is first fed into an ECA layer. Within this layer, the aforementioned adaptive 1D convolution is performed, and a Sigmoid activation function is applied to compute an attention weight scalar in the [0, 1] range for each channel. Subsequently, these attention weights are multiplied channel-wise with the original feature vector $z_{i}$ to obtain a channel-recalibrated feature vector ${z^{\prime }}_{i}$. Finally, the recalibrated vector ${z^{\prime }}_{i}$ is passed through a standard fully connected layer, which directly maps it to the final output vector ${\hat{y}}_{i}$, containing the two predicted targets (total length and weight).

This ECA + Linear regression head design constitutes a computationally efficient prediction module. Under the premise of introducing almost no additional computational overhead, it endows the model with the ability to dynamically focus on key feature channels. This capability is particularly crucial for a model learning task-specific effective feature representations from scratch, as it helps accelerate model convergence and improve the discriminability of the learned features.

### 2.4. Design of the Hybrid Loss Function

We selected the mean absolute error (MAE), also known as L1 loss, as the primary regression loss term. Compared to the mean squared error (MSE) loss, L1 loss offers several significant advantages. The MSE loss, which calculates the mean of the squared errors between predicted and true values, tends to amplify the influence of outliers, making the model training susceptible to interference from extreme data points. In contrast, the L1 loss uses the mean of the absolute errors, exhibiting greater robustness to outliers and effectively preventing significant performance degradation when handling datasets containing noise or outliers.

Furthermore, the gradient of the L1 loss is a constant value, which contributes to more stable training, particularly in the initial stages, and reduces the risk of exploding gradients. Its optimization process also tends to produce sparse solutions, which can implicitly perform feature selection. Conversely, the gradient of the MSE loss grows linearly with the error, which can lead to instability during the early phases of training. Additionally, the physical interpretation of the L1 loss is more intuitive, as it directly corresponds to the mean absolute error, making it easier to understand the model’s prediction bias. Therefore, in regression tasks where robustness, sparsity, or sensitivity to absolute error is prioritized, L1 loss is the superior choice. It is defined as follows:(6)$$L_{R}=\frac{1}{N}\overset{N}{\underset{i=1}{\sum }}\left(\left|l_{i}\;-\;{\hat{l}}_{i}\right|+\left|w_{i}-{\hat{w}}_{i}\right|\right)$$
where N is the number of samples in a batch, and $y_{i}$ and ${\hat{y}}_{i}$ are the ground truth and predicted output vectors for the i−th sample, respectively.

To compensate for the shortcomings of the primary regression loss and enhance the model’s ability to handle imbalanced data, we introduced the adaptive contrastive regularization loss, denoted as $L_{\mathrm{sum}}$, as a regularization term. By incorporating $L_{\mathrm{sum}}$, we not only require the model’s predicted values to approach the ground truth but also impose a strong constraint at the feature space level. The model must learn a structured feature representation capable of distinguishing between samples with different size biases. This mechanism, through its contrastive learning strategy of “pulling similar samples closer and pushing dissimilar ones apart,” effectively prevents minority class samples from being marginalized in the feature space. This, in turn, significantly improves the model’s generalization ability and predictive fairness across the entire sample space.

Ultimately, the model’s optimization objective is to minimize the total loss function, $L_{\mathrm{sum}}$, which is a weighted sum of the L1 loss and the AMCR loss:(7)$$L_{\mathrm{sum}}=L_{R}+\beta L_{\mathrm{AMCR}}$$
where β is a hyperparameter used to balance the importance of the primary regression task and the contrastive regularization task. Guided by this hybrid loss function, the model simultaneously optimizes for two goals: predictive accuracy and feature space structure, thereby synergistically enhancing overall performance.

### 2.5. Model Compression and Fine-Tuning

To enhance the model’s deployment efficiency on resource-constrained edge devices, we further explored a model lightweighting pipeline after the initial training phase was completed. We employed a structured pruning technique, which directly reduces the model’s parameter count and computational complexity by physically removing entire structural units from the network.

#### 2.5.1. General Structured Pruning Based on Dependency Graphs

To address the prevalent issue of structural coupling in neural networks—where parameters in different layers are interdependent due to network connections and must be pruned simultaneously—and to enable general-purpose structured pruning for arbitrary network architectures, this study adopted the advanced dependency graph (DepGraph) method [37], utilizing the implementation provided in the Torch-Pruning library (v1.5.2). The core idea of DepGraph is to automatically model the complex dependency relationships within a network. Its key steps and innovations are as follows:

Network decomposition and dependency graph construction: DepGraph first decomposes the network into fine-grained basic components, with each component representing a parameterized layer or a non-parameterized operation. Each component, $f_{i}$, is then further resolved into an input node, $f_{i}^{-}$, and an output node, $f_{i}^{+}$. Based on this representation, DepGraph automatically constructs a dependency graph. Inter-layer dependency: If the output of component $f_{i}$ is directly connected to the input of component $f_{j}$, a dependency, $f_{i}^{+}\to f_{j}^{-}$, exists. Intra-layer dependency: If the input and output of a component, fᵢ, share the same pruning scheme, $sch\left(f_{i}^{-}\right)=sch\left(f_{i}^{+}\right)$, a dependency, $f_{i}^{-}\Leftrightarrow f_{i}^{+}$, exists. The dependency graph D only records the direct dependencies between adjacent layers or within a layer’s input/output nodes. It serves as a transitive reduction in the original dense grouping matrix G, containing the same information but with the minimum number of edges. As illustrated in Figure 6a, a network segment containing a convolutional layer and a BN layer is decomposed into input $f_{i}$ and output $f_{j}$ nodes, and a dependency graph comprising both inter- and intra-layer dependencies is constructed.

Automatic identification of dependency groups: Based on the constructed dependency graph D, DepGraph identifies all maximal connected subgraphs using a graph traversal algorithm. Each connected subgraph constitutes an indivisible dependency group. As shown in Figure 6a under “Dependency Group Identification,” through graph traversal, all nodes in the dependency graph are identified as a single pruning group due to their connectivity. A pruning operation on any parameter within a group will have its effects automatically, synchronously, and safely propagated to all other related parameters in the same group via these dependency relationships. This process rigorously ensures the structural integrity and functional validity of the network after pruning.

Traditional methods assess importance at the single-layer or parameter level, making it difficult to accurately measure the overall contribution of a structurally coupled dependency group. DepGraph addresses this issue by applying sparsity training at the dependency group level. This is achieved by introducing a group-level L2-norm regularization term:(8)$$R\left(g,k\right)=\underset{k}{\sum }\gamma_{k}\cdot (\underset{w\in g}{\sum }\|w\left[k\right]{\|}_{2}^{2})$$
where k indexes the prunable dimensions within group g. An adaptive shrinkage strength, $\gamma_{k}$, is employed, defined as $\gamma_{k}=2^{\alpha \left(I_{g}^{\max}\;-I_{g,k}\right)/\left(I_{g}^{\max}\;-I_{g}^{\min}\right)}$, where α is a hyperparameter. This strategy applies stronger shrinkage to dimensions of lower importance, thereby learning a consistent sparsity pattern within the group.

After sparsity training, dimensions of low importance within a group are sparsified to zero and can thus be safely removed, a process illustrated in Figure 6b. This mechanism enables effective group-level pruning, even with simple norm-based criteria. The complete process is depicted in Figure 6, which clearly demonstrates how to start from a network block, construct the dependency graph of its internal components, and subsequently perform sparsity training on the resulting dependency group to ultimately remove redundant dimensions and achieve model compression.

#### 2.5.2. Pruning and Fine-Tuning Pipeline

Based on the DepGraph methodology, we executed the following: “pruning and fine-tuning” pipeline on our trained largemouth bass prediction model:Dependency-graph construction and group identification: The well-trained, high-accuracy bass prediction model, along with a sample input, was fed into the torch-pruning framework. The framework automatically performed network decomposition, constructed the dependency graph D, and identified all indivisible dependency groups through graph traversal.Group-level sparsity training: The model underwent a short period of retraining on the original training set, with group-level sparsity regularization applied. This step learned a consistent sparsity mask for each dimension within every dependency group, thereby identifying redundant dimensions.Pruning execution: guided by the sparsity masks learned during training, all dimensions or entire dependency groups identified as redundant were iteratively removed until a predefined global pruning ratio was achieved. Key safeguard: to preserve the core prediction functionality, the final linear layer of the regression head was explicitly excluded from the pruning scope.Fine-tuning: Structured pruning inevitably causes a temporary degradation in model performance. After the pruning was completed, the compacted model was fine-tuned on the original training set. This fine-tuning process involved training for a limited number of epochs with a small learning rate. The objective was to help the model recover and optimize its predictive accuracy on the new, pruned architecture, ultimately aiming to match or even surpass its pre-pruning performance level.

## 3. Results

### 3.1. Experimental Setup and Evaluation Metrics

To ensure the scientific rigor, objectivity, and reproducibility of our research findings, this section provides a detailed account of the hardware and software environments, data processing pipeline, specific parameters for model training and evaluation, and the metrics used to assess model performance.

#### 3.1.1. Experimental Environment and Hyperparameter Settings

The experiments in this study were conducted in the following hardware and software environment. The hardware platform was a workstation equipped with an Intel Core i5-12600KF CPU, 32 GB of DDR4 memory, and an NVIDIA GeForce RTX 4060 Ti GPU. High-performance NVMe SSDs were used for system storage. The operating system was Windows 10 Professional.

The software environment was built upon the Python v3.8.19 programming language. The deep learning framework employed was PyTorch v1.13.1 with CUDA v11.7 acceleration enabled. The key dependent libraries included the following: OpenCV v4.10.0 and TorchVision v0.14.1 for computer vision tasks; Torch-Pruning v1.5.2 for model compression; NumPy v1.24.3 and SciPy v1.10.1 for scientific computing; and Pandas v1.5.3 for data manipulation. The GPU acceleration environment was facilitated by the CUDA Toolkit v11.7 and cuDNN v8.5.0, with versions strictly aligned with the PyTorch framework to ensure computational stability. The baseline hyperparameter settings are detailed in Table 3.

#### 3.1.2. Evaluation Metrics

To comprehensively evaluate the model’s performance on the bass length and weight prediction tasks, this study employed the following four widely used regression evaluation metrics:Root mean square error (RMSE): $\mathrm{RMSE}=\sqrt{\frac{1}{N}\sum_{i=1}^{N}{\left(y_{i}-{\hat{y}}_{i}\right)}^{2}}$. This metric measures the degree of dispersion between predicted and true values and is more sensitive to larger errors. A lower RMSE value indicates higher prediction accuracy.Mean absolute error (MAE): $\mathrm{MAE}=\frac{1}{N}\sum_{i=1}^{N}|y_{i}-{\hat{y}}_{i}|$. This metric reflects the average absolute magnitude of the prediction error and is less sensitive to outliers. It provides a direct representation of the actual level of prediction error.Mean absolute percentage error (MAPE): $\mathrm{MAPE}=\frac{100\%}{N}\sum_{i=1}^{N}\left|\frac{y_{i}-{\hat{y}}_{i}}{y_{i}}\right|$. This metric quantifies the relative error in percentage form, eliminating the influence of scale and facilitating comparisons across prediction tasks with different dimensions.Coefficient of determination (R^2^): $R^{2}=1-\frac{\sum_{i=1}^{N}{\left(y_{i}-{\hat{y}}_{i}\right)}^{2}}{\sum_{i=1}^{N}{\left(y_{i}-\bar{y}\right)}^{2}}$. This metric indicates the explanatory power of the model for the variance in the target variable. Its value ranges from 0 to 1, with a value closer to 1 signifying a better model fit.

### 3.2. Comparative Experiments

To systematically validate the effectiveness of our ECR-MobileNet model, which integrates ECA and AMCR, we conducted a benchmark study against 14 mainstream lightweight models. The selected models, including classic CNNs, hybrid architectures, and vision transformers, were chosen based on their best-performing variants with a parameter count under 25 M. To ensure a fair comparison, all models were trained from scratch on the largemouth bass dataset and evaluated on an independent test set, under the unified experimental environment described in Section 3.1.1.

#### 3.2.1. Performance Analysis of Length and Weight Prediction

We evaluated our proposed models against 14 mainstream lightweight architectures on the largemouth bass biometric prediction task. The results, summarized in Table 4 and Table 5, demonstrate the superior accuracy of our approach.

For length prediction (Table 4), our unpruned ECR-MobileNet achieved a highly competitive RMSE of 0.4565 and the lowest MAE (0.2149) and MAPE (0.93%) among all baseline models. Crucially, the pruned ECR-MobileNet-P model further improved performance, attaining a state-of-the-art RMSE of 0.4296 and an R^2^ of 0.9784. This result shows a 3.6% RMSE reduction compared to the best-performing baseline, ShuffleNetV2-x1.0, highlighting that our pruning strategy enhances generalization, rather than degrading accuracy.

The performance advantage was even more significant in the more challenging weight prediction task (Table 5). The ECR-MobileNet-P model ranked first across all four metrics, achieving an RMSE of 0.0202, MAE of 0.0108, MAPE of 3.31%, and an R^2^ of 0.9740. This outcome indicates that our integrated framework, combining the ECA module, AMCR loss, and structured pruning, effectively models the complex, non-linear relationship between fish morphology and weight, setting a new performance benchmark for this task.

#### 3.2.2. Comprehensive Evaluation

A comprehensive analysis of the results reveals the consistent cross-task superiority of the proposed ECR-MobileNet framework. In both length and weight prediction tasks, our models systematically outperformed the 14 baseline models, which include classic CNNs, hybrid architectures, and vision transformers.

Notably, the pruned ECR-MobileNet-P consistently delivered the best overall performance, demonstrating that our “pruning as optimization” pipeline is highly effective. This approach not only reduces model complexity but also acts as a powerful regularizer, refining feature representations to enhance predictive accuracy. The consistent top-ranking performance across two distinct but related biological prediction tasks validates the robustness and generalizability of our integrated design, which synergistically combines attention mechanisms ECA, imbalanced regression techniques AMCR, and structured pruning. This establishes a new state-of-the-art performance benchmark in non-contact fish biometric assessment.

### 3.3. Analysis of Model Compression and Fine-Tuning Results

To evaluate the performance of our proposed ECR-MobileNet model and its pruned version in terms of computational efficiency and deployment feasibility, we conducted a comprehensive quantitative analysis of their model complexity. This analysis was benchmarked against a series of mainstream models, using metrics that include parameter count (M), computational load (GFLOPs), single-core CPU inference latency (ms), and memory footprint (MB).

#### 3.3.1. Inherent Efficiency of ECR-MobileNet

First, even before pruning, our proposed ECR-MobileNet model already demonstrated exceptional efficiency. As detailed in Table 6, ECR-MobileNet has a parameter count of only 0.93 M and a computational load of just 0.12 GFLOPs. This complexity is substantially lower than that of most baseline models. Its computational load is comparable to that of MobileNetV3-Small but is significantly lower than that of ShuffleNetV2-x1.0 (0.30 GFLOPs). This inherent efficiency is primarily attributed to its lightweight MobileNetV3-Small backbone and our custom-designed ECA regression head, which introduces an attention mechanism with almost no additional computational burden.

#### 3.3.2. Extreme Lightweighting Achieved Through Structured Pruning

A core contribution of this research is the validation of the significant potential of advanced structured pruning techniques in achieving extreme model lightweighting. By adopting the dependency-graph-based pruning strategy combined with fine-tuning, we obtained the ECR-MobileNet-P model. This model demonstrated remarkable optimization in its efficiency metrics. Its parameter count was reduced to 0.52 M, and its computational load was a mere 0.07 GFLOPs. Compared to the unpruned ECR-MobileNet, the parameter count and computational load were reduced by 44.1% and 41.7%, respectively. When benchmarked against ShuffleNetV2-x1.0, the most efficient of the baseline models, the pruned model’s parameter count was only 41.6% of the latter’s, and its computational load dropped to an impressive 23.3%.

This enhancement in efficiency directly translates to faster real-world inference speeds. The CPU latency of ECR-MobileNet-P was only 10.19 ms, making it 29.5% faster than its unpruned counterpart and superior to all baseline models. Concurrently, its memory footprint was compressed to 2.00 MB, the lowest among all models, which substantially lowers the hardware threshold for deployment.

#### 3.3.3. Comprehensive Evaluation of Performance and Efficiency

Synthesizing the performance analysis from Section 3.2, our proposed ECR-MobileNet-P model achieves a Pareto-optimal state, excelling in both the performance and efficiency dimensions. The model outperforms all 14 baseline models in prediction accuracy for both fish length and weight, with the minor exception of MAE and MAPE in length prediction, where it is marginally surpassed by the unpruned ECR-MobileNet. Simultaneously, it secures the top rank across all four key efficiency metrics: parameter count, computational load, inference latency, and memory footprint.

This achievement holds significant practical value, as it fully validates the effectiveness of the “advanced model architecture (ECA+AMCR) + general structured pruning” technical pipeline. Not only did we design a base model that combines high accuracy with inherent efficiency, but we also, through a subsequent automated pruning process, pushed its efficiency to the extreme while maintaining or even enhancing its predictive accuracy. This work provides a solid technical foundation and a viable solution for developing real-time, high-precision fish phenotyping systems intended for either resource-constrained embedded devices or high-throughput online servers.

### 3.4. Ablation Studies

To rigorously evaluate the independent contributions of each innovative technique proposed in this study, as well as their synergistic effects, we designed and conducted a series of detailed ablation studies. We established a baseline model comprising a MobileNetV3-Small backbone, a simple linear regression head, and training with the standard MSE loss function. We then progressively introduced or replaced components with our proposed enhancements and evaluated the performance of all model combinations on both the length and weight prediction tasks.

#### 3.4.1. Ablation Analysis on the Length Prediction Task

We conducted detailed ablation studies to dissect the contribution of each component within our ECR-MobileNet framework. The results for the length prediction task are presented in Table 7.

The analysis demonstrates that each proposed module—the L1 loss, the efficient channel attention (ECA) module, and the adaptive multi-scale contrastive regression (AMCR) loss—independently improved model performance over the baseline. For instance, replacing the standard MSE with L1 loss enhanced robustness to outliers (lower MAE/MAPE), while integrating the ECA module improved feature discriminability, leading to a reduction in all error metrics.

More importantly, the studies revealed a strong synergistic effect among the components. The combination of L1 loss and the ECA module (ECA+L1) yielded significant improvements beyond their individual contributions, achieving an RMSE of 0.4951. The full model, integrating all three components, achieved the global optimum across all metrics (RMSE: 0.4565, R^2^: 0.9757). This confirms that our design is not a simple aggregation of techniques, but a cohesive system where the robustness of L1 loss, the feature enhancement of ECA, and the imbalance-resistance of AMCR work in concert to achieve state-of-the-art performance. A similar synergistic pattern was observed in the weight prediction task (Table 8), further validating the stability and effectiveness of our proposed framework.

#### 3.4.2. Ablation Analysis on the Weight Prediction Task

On the weight prediction task, we observed a pattern highly consistent with that of the length task, further validating the generalizability and stability of our proposed methods. As shown in Table 8, the independent contributions of L1 loss and the ECA module were once again confirmed. Introducing either L1 loss or the ECA module alone yielded significant performance improvements; notably, the ECA module substantially reduced MAPE from 4.61 to 4.22%. The ECA+L1 combination performed exceptionally well, achieving an RMSE of 0.0233 and a MAPE of 3.55%, making it the best-performing of all pairwise combinations and proving the immense power of integrating structural optimization with a robust loss function.

Ultimately, the full model integrating all three enhancements once again achieved an overwhelming victory, securing the global best for all metrics. It particularly excelled by lowering the MAE to 0.0109 and the MAPE to 3.50%. This result reconfirms that our proposed trinity of optimization strategies is the optimal solution for achieving high-precision weight prediction.

Based on the systematic ablation studies described above, we draw the following conclusions: Each enhancement proposed in this study—replacing MSE with the more robust L1 loss, integrating the efficient ECA module into the regression head, and introducing the AMCR loss designed for imbalanced data—makes an independent, positive contribution to the final model performance. More importantly, these techniques are not a simple aggregation but, rather, mutually reinforcing, creating a synergistic effect that ultimately enabled our complete model to achieve state-of-the-art (SOTA) performance on both tasks.

### 3.5. Visualization Analysis

To provide a more intuitive demonstration of the training stability of our proposed ECR-MobileNet model and the superior performance of its pruned version, ECR-MobileNet-P, we conducted a series of visualization analyses. These analyses include the evolution of the loss function during the training process, the fit between predicted and true values on the test set for the final models, and a comparison of error distributions against the baseline model.

#### 3.5.1. Analysis of the Training Process

Figure 7 presents the loss function curves of the ECR-MobileNet model over 200 training epochs, comprising four key trajectories: Total Training Loss, Validation L1 Loss, Train L1 Loss Component, and Train AMCR Loss Component.

Several key observations can be drawn from the figure:

Stable model convergence: The total training loss curve (solid dark blue line) exhibits a smooth, monotonically decreasing trend. The rapid decay in the initial phase indicates that the model efficiently captures the fundamental patterns in the data, while the gradual convergence to a stable range in the later stages validates the effectiveness and stability of the training process.

Assured generalization ability: The validation L1 loss (solid pink line) and the training L1 loss component (dashed purple line) consistently maintain a highly synchronized downward trend, with no signs of the validation loss increasing at any point. This confirms the efficacy of our data augmentation strategies and the AMCR regularization, which significantly suppresses the risk of overfitting and ensures the model’s generalization performance.

Synergistic regularization of AMCR: The AMCR loss component (dotted orange line) decreases steadily, with its optimization trajectory highly coordinated with that of the main loss function. This indicates that the contrastive regularization term continuously guides the structuring of the feature space. Through joint optimization with the regression task, it drives the model to converge towards a more discriminative solution space.

#### 3.5.2. Visualization of Prediction Performance

To intuitively assess the prediction accuracy of the pruned ECR-MobileNet-P model, Figure 8 displays scatter plots of the predicted versus true values for both body length and weight on the test set. In the plots, the horizontal and vertical axes represent the true and predicted values, respectively, while the dashed black line indicates the ideal fit reference, y = x.

Figure 8a illustrates the results for length prediction. The data points are tightly clustered along the y = x reference line, exhibiting a strong linear trend. The coefficient of determination reached 0.9784, visually confirming the high degree of agreement between the predicted and true values. Figure 8b shows the results for weight prediction, where the data points also form a compact linear distribution, with an R^2^ value of 0.9740. This further substantiates the model’s high-precision fitting capability on the weight prediction task.

Collectively, both plots corroborate the quantitative conclusions from Section 3.2: ECR-MobileNet-P achieves high correlation with the ground truth and demonstrates exceptional regression accuracy on both the length and weight prediction tasks.

#### 3.5.3. Comparative Analysis of Error Distributions

To quantitatively assess the performance advantage of the proposed method over the baseline model, Figure 9 presents histograms of the absolute prediction errors on the test set.

Figure 9a,b display the absolute error distributions for the length and weight tasks, respectively. In both prediction tasks, the frequency of ECR-MobileNet-P (orange) in the low-error intervals (length error < 0.2 cm, weight error < 0.01 kg) is significantly higher than that of the baseline model (blue). This indicates that its predictions exhibit a superior concentration of accuracy. Conversely, the baseline model shows a distinct heavy-tail phenomenon in the high-error intervals (length error > 1.0 cm, weight error > 0.04 kg), whereas the tail of the ECR-MobileNet-P distribution rapidly converges to near-zero frequency, effectively suppressing the occurrence of extreme prediction errors.

This difference in distribution morphology demonstrates that the proposed ECR-MobileNet-P not only reduces the systematic bias (indicated by the leftward shift of the distribution) but also decreases the prediction variance (evidenced by the increased kurtosis of the distribution). This enables the model to exhibit stronger robustness on complex samples.

## 4. Discussion

By proposing the ECR-MobileNet-P model, this study successfully addresses the problem of non-contact prediction for the body length and weight of largemouth bass, providing an efficient and accurate solution for intelligent aquaculture. In the following sections, we discuss the interpretation of our results, compare them with the existing literature, and elaborate on the implications for aquaculture, the limitations of our study, and future research directions to comprehensively evaluate the contributions and potential impact of our work.

### 4.1. Interpretation of Results

The experimental results demonstrate that the ECR-MobileNet-P model achieved state-of-the-art performance on both the length and weight prediction tasks. Specifically, the model achieved an RMSE of 0.4296 cm, an MAE of 0.2310 cm, a MAPE of 0.99%, and an R^2^ of 0.9784 for length prediction. For weight prediction, it achieved an RMSE of 0.0202 kg, an MAE of 0.0108 kg, a MAPE of 3.31%, and an R^2^ of 0.9740. These metrics surpassed those of all 14 baseline models, which included various types of lightweight deep learning architectures such as EfficientNetV2, GhostV3, and vision transformers, covering both convolutional neural network and vision transformer paradigms (see Section 3.2 for details). This level of accuracy, which significantly reduces the typical 5–10% error margin of manual methods, establishes a new state-of-the-art benchmark for non-contact fish biometrics and addresses a key challenge in deploying precise computer vision systems in real-world aquaculture.

The superior performance of ECR-MobileNet-P can be attributed to the following key designs:ECA mechanism: By adaptively adjusting channel weights, the ECA module [36] enhanced the model’s ability to focus on critical features, which is crucial for distinguishing salient morphological landmarks from a cluttered and dynamic underwater background.AMCR loss function: Through its contrastive learning strategy, AMCR dynamically constructed positive and negative pair relationships among samples, effectively mitigating the data imbalance problem and ensuring fair predictions across different fish sizes.Structured pruning: By employing the dependency-graph-based DepGraph pruning technique [37], the parameter count of ECR-MobileNet-P was reduced from 0.93 M to 0.52 M, and the computational load from 0.12 GFLOPs to 0.07 GFLOPs—a reduction in 44.1% and 41.7%, respectively. This method is particularly suitable for compact architectures like MobileNetV3, as it correctly identifies and preserves complex inter-layer dependencies, preventing structural damage that simpler pruning methods might cause. After fine-tuning, the model’s performance not only recovered but surpassed that of the original model, validating the potential of pruning as an optimization tool.

The ablation studies further quantified the contribution of each component. The combination of the ECA module and the AMCR loss function significantly enhanced model performance, while the pruning technique acted as a form of regularization by reducing redundant parameters, further refining the feature representations (see Section 3.4 for details). The model’s CPU inference latency of just 10.19 ms and its low memory footprint of 2.00 MB—far below other lightweight models—demonstrate its practical utility in resource-constrained environments.

### 4.2. Comparison with Existing Literature

The present study represents a significant advancement over the existing body of literature. Previous research on fish biometric prediction has primarily focused on either traditional machine learning methods or computationally intensive deep learning models. For instance, Zhang et al. [28] employed statistical morphological analysis and machine learning to estimate tilapia biomass. While their approach achieved high accuracy, it was constrained by its reliance on manually extracted features, resulting in poor generalization. In contrast, our study leverages deep learning for automatic feature extraction, making it applicable to a broader range of species and environments, and it particularly excels in its robustness under complex underwater conditions.

A review by Saleh et al. [38] on the application of computer vision and deep learning for underwater fish classification highlighted the challenges that complex aquatic environments pose to model performance. Our work addresses this specific gap by utilizing a multi-scene dataset and robust data augmentation strategies, which significantly enhanced the model’s robustness in such challenging environments. Unlike the studies by Jansi Rani et al. [29] and Yan et al. [30], our research places a dual emphasis on both prediction accuracy and computational efficiency, making it suitable for deployment on edge devices in small-scale aquaculture farms. Furthermore, while the review by Yan et al. [30] underscored the sustainable impact of AI in aquaculture, it did not delve into the practical deployment challenges of lightweight models. Our work directly confronts this issue. By combining structured pruning with the ECA module, we achieved substantial reductions in parameter count and computational load, thus addressing a critical void in the literature regarding the practical implementation of efficient models. Furthermore, our work extends the findings of recent studies on imbalanced regression. For instance, while Yang et al. [35] laid the theoretical groundwork for deep imbalanced regression, their methods were primarily tested on general-purpose datasets. Our study is one of the first to successfully adapt and extend these principles, specifically through our AMCR loss, to a complex, real-world biological application with continuous, multi-dimensional targets. Similarly, our work builds upon the non-contact measurement systems reviewed by Li et al. [6] and Saleh et al. [38] by explicitly tackling the dual challenges of data imbalance and model lightweighting, which were identified as key bottlenecks for practical deployment.

### 4.3. Implications for Aquaculture

The successful development of the ECR-MobileNet-P model provides the aquaculture industry with an efficient, accurate, and non-invasive tool for growth monitoring. Traditional manual measurement methods are not only labor-intensive but also induce stress in fish, adversely affecting their health and overall farming efficiency. In contrast, the model proposed in this study enables non-destructive, real-time monitoring through computer vision, which can significantly reduce manual intervention and fish stress while simultaneously improving the accuracy and efficiency of data acquisition.

By providing real-time predictions of fish length and weight, aquaculture farms can more precisely calculate feed requirements, thereby reducing waste and enhancing economic returns [5]. Accurate growth parameter prediction also facilitates more precise biomass estimation, which supports production planning and market-making decisions at the farm level. The non-contact nature of the measurement reduces physical disturbance to the fish, lowering stress levels and promoting sustainable aquaculture practices [38]. Furthermore, owing to the lightweight design of ECR-MobileNet-P, the model can be deployed on edge devices at small-scale farms without relying on high-performance servers. This substantially lowers the barrier to technology adoption and offers a viable intelligent solution for small and medium-sized aquaculture enterprises [39].

### 4.4. Limitations

Despite the significant achievements of this study, several limitations should be acknowledged:The current dataset was primarily collected from specific aquaculture environments in Hunan Province, which may not fully represent the diverse farming conditions (e.g., water quality, illumination) found in different regions globally. Therefore, the model’s generalization ability in other environments requires further validation.Although this study enhanced the model’s robustness through a multi-scene dataset, extreme conditions—such as severely turbid water or rapidly swimming fish—may still affect prediction accuracy. This necessitates further testing under such challenging scenarios.The pruned model exhibited a marginal decrease in certain metrics compared to its unpruned counterpart. While its overall performance remained superior to the baseline models, this trade-off between efficiency and performance must be carefully considered in practical applications.

### 4.5. Future Research Directions

This research opens up several avenues for future work:Validating the model’s performance on different fish species and across diverse geographical regions to assess its generalization capabilities [5].Exploring techniques such as quantization, neural architecture search (NAS), or hybrid CNN-Transformer architectures to further reduce the model’s computational demands while maintaining or even enhancing prediction accuracy [35].Incorporating generative adversarial networks (GANs) [40] or self-supervised learning methods [41] to further bolster the model’s adaptability in complex and varied environments.

### 4.6. Theoretical and Methodological Contributions

This study makes the following contributions to the field of deep learning:AMCR provides a novel solution to the data imbalance problem in regression tasks by dynamically constructing positive and negative pair relationships among samples. Its methodology is potentially generalizable to other domains where imbalanced regression is a challenge [35].This research demonstrates that pruning can serve as a form of regularization by reducing redundant parameters, not merely as a tool for reducing model complexity. This finding challenges the conventional notion that pruning inevitably sacrifices accuracy, thereby offering new insights for model compression research.

In conclusion, through innovative model design, loss function optimization, and advanced pruning techniques, this study has developed a fish biometric prediction model that achieves a leading edge in both accuracy and efficiency. ECR-MobileNet-P not only provides a practical tool for intelligent aquaculture but also forges a new path for the application of deep learning in resource-constrained environments. In the future, with further refinement and broader adoption, the proposed model is poised to play a significant role in a wider range of applications, driving the intelligent and sustainable development of the aquaculture industry.

## 5. Conclusions

This study successfully developed ECR-MobileNet-P, a lightweight and robust deep learning framework for non-contact, high-precision prediction of largemouth bass length and weight. By synergistically integrating an ECA module, a novel AMCR loss for imbalanced data, and a dependency-graph-based structured pruning pipeline, our model effectively addresses the dual challenges of prediction accuracy and deployment efficiency in aquaculture.

Our final pruned model, ECR-MobileNet-P, achieved state-of-the-art performance, outperforming 14 mainstream models with an RMSE of 0.4296 for length and 0.0202 for weight, while maintaining a minimal computational footprint (0.52 M parameters, 10.19 ms CPU latency). This work makes two key contributions: first, it provides a practical, scalable, and stress-free tool for intelligent fish growth monitoring; second, it establishes an effective “pruning as optimization” paradigm, demonstrating that structured pruning can simultaneously enhance model efficiency and generalization.

Future research will focus on three primary directions. First, we will expand the dataset to include more fish species and diverse geographical environments to validate and improve the model’s generalization capabilities. Second, we will explore advanced compression techniques, such as quantization and neural architecture search (NAS), to enable deployment on ultra-low-power edge devices. Finally, we aim to integrate this biometric prediction module into a multi-modal system that also analyzes fish behavior and health status, creating a holistic decision support tool for precision aquaculture.

## Figures and Tables

**Figure 1 animals-15-02443-f001:**
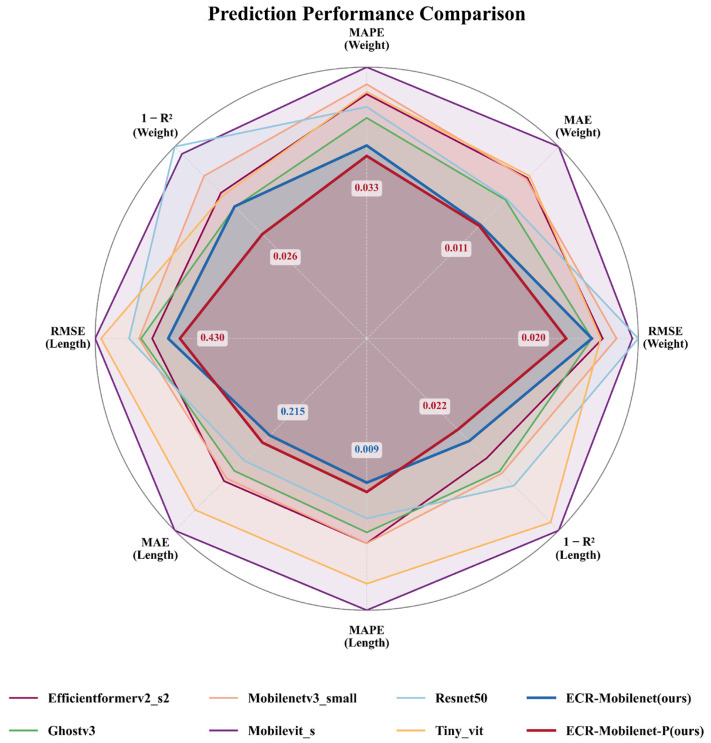
Radar chart comparing the performance of various models on the largemouth bass body length and weight prediction task. The chart illustrates the comparison across eight key regression metrics, including the mean absolute error (MAE), mean absolute percentage error (MAPE), root mean square error (RMSE), and 1 − R^2^. For all metrics, lower values indicate superior performance; consequently, a smaller polygon area closer to the center signifies better comprehensive performance. The results demonstrate that our proposed ECR-MobileNet (blue) significantly outperforms all other comparative models across every metric, validating the effectiveness of its design (Contribution 1). Furthermore, its pruned version, ECR-MobileNet-P (red), achieves even greater performance while drastically reducing model complexity, convincingly demonstrating the success of our proposed lightweighting pipeline (Contribution 2).

**Figure 2 animals-15-02443-f002:**
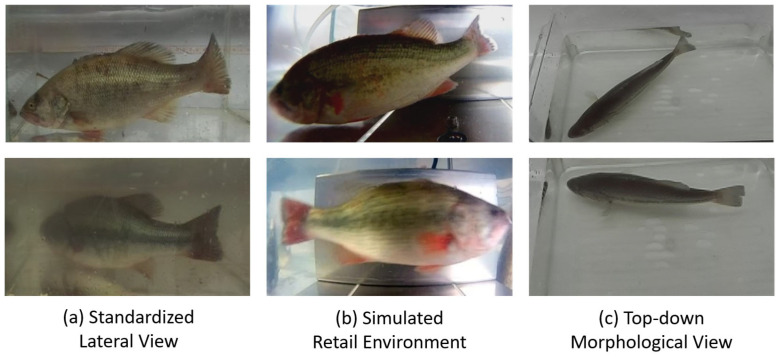
Representative images from the multi-scene largemouth bass dataset, designed to address key challenges in the aquaculture industry chain. (**a**) The left column displays standardized lateral-view measurement scenarios, with samples in clear (**top**) and simulated turbid (**bottom**) water to test the model’s environmental adaptability. (**b**) The middle column presents simulated flow-through environment monitoring scenarios, featuring challenges such as variable lighting conditions and complex non-rigid poses. (**c**) The right column shows top-down morphometric measurement scenarios for capturing body width and dorsal contour information, which is crucial for accurate biomass estimation.

**Figure 3 animals-15-02443-f003:**
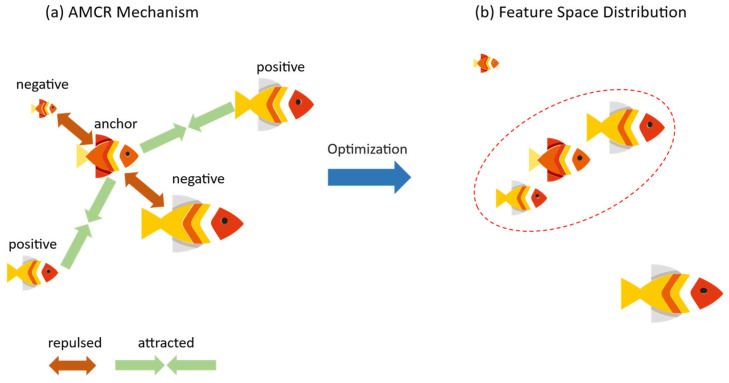
Conceptual diagram of the core principle behind adaptive multi-scale contrastive regression (AMCR). (**a**) The contrastive learning mechanism: centered on an anchor sample, the model is trained to attract positive samples with similar biometric features (Attraction) and repel negative samples with dissimilar features (Repulsion) in the feature space. (**b**) The desired feature space distribution after training: Following optimization with AMCR, samples with similar biometric characteristics (e.g., fish of comparable size) form compact clusters.

**Figure 5 animals-15-02443-f005:**
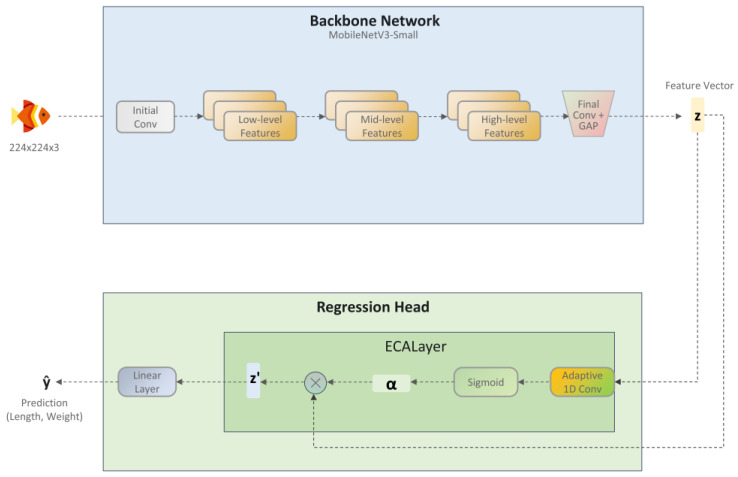
The overall architecture of the proposed ECR-MobileNet. The model comprises two main components: (1) A lightweight backbone network based on MobileNetV3-Small, which performs hierarchical feature extraction to output a feature vector z. (2) An efficient regression head, which first uses an ECA layer to recalibrate the channel-wise importance of z (yielding z’) and then employs a final linear layer to predict the biometric parameters (i.e., total length and weight).

**Figure 6 animals-15-02443-f006:**
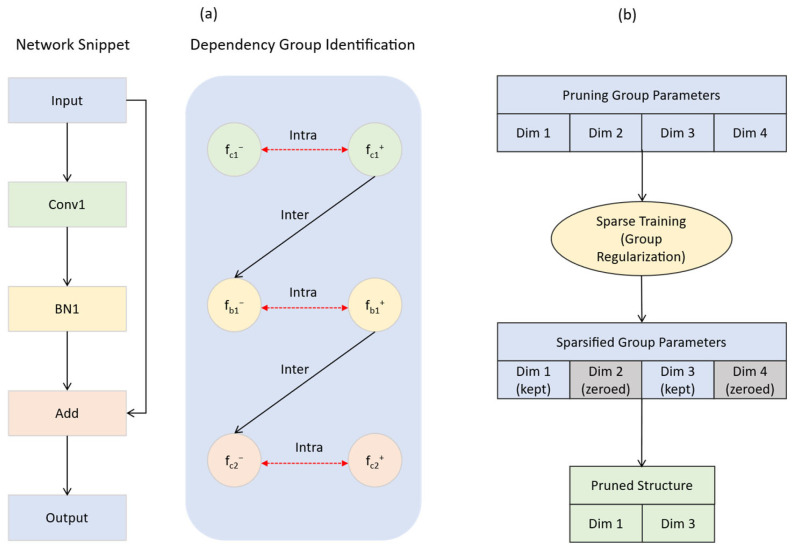
Diagram of the structured pruning process based on a dependency graph (DepGraph). (**a**) The first stage involves decomposition and dependency graph construction, where an original network module is abstracted to identify both intra-layer and inter-layer parameter dependency groups. (**b**) The second stage is group-level sparse training, where group sparsity regularization is applied to drive the weights of entire redundant dimensions to zero, leading to a final pruned structure. This entire process achieves model compression while ensuring structural integrity.

**Figure 7 animals-15-02443-f007:**
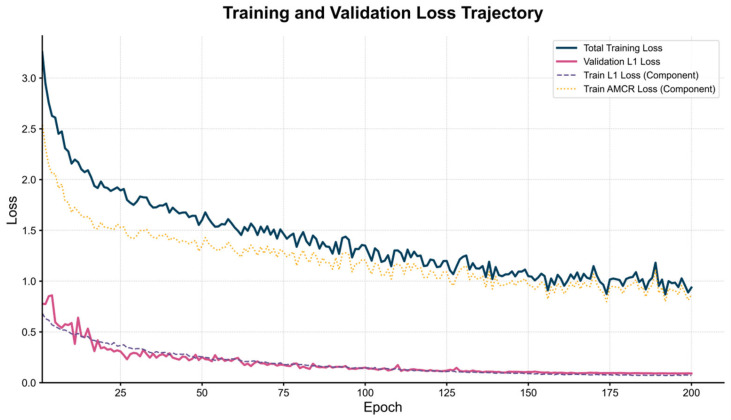
Training and validation loss trajectories for the ECR-MobileNet model. The figure illustrates the loss dynamics over 200 training epochs. The total training loss (solid dark blue line) smoothly decreases, indicating stable model convergence. The validation L1 loss (solid pink line) and the training L1 loss component (dashed purple line) maintain a synchronized descent throughout the process, showing no signs of overfitting and thus demonstrating the model’s strong generalization ability.

**Figure 8 animals-15-02443-f008:**
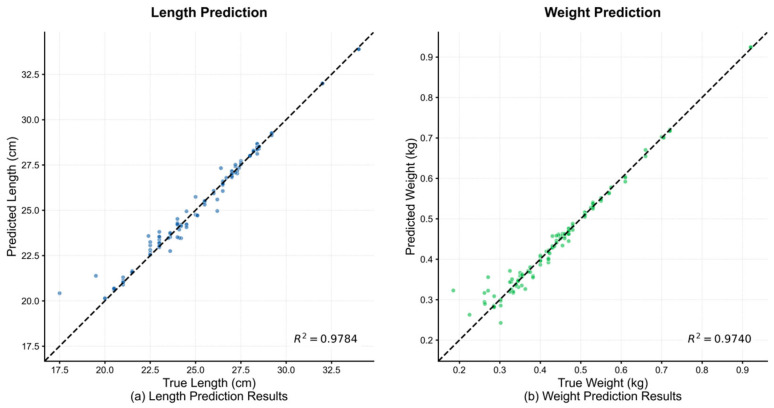
Performance visualization of the ECR-MobileNet-P model on the length and weight prediction tasks. The scatter plots clearly show that, for both tasks, the predicted values exhibit a high degree of linear agreement with the true values, as the data points are tightly clustered around the ideal reference line, y = x.

**Figure 9 animals-15-02443-f009:**
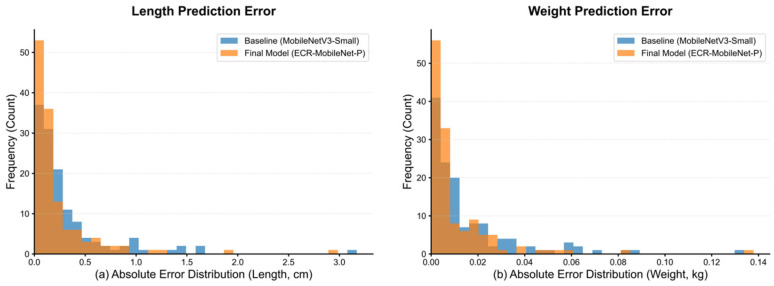
Histograms of prediction error distributions for the final and baseline models. The figure compares the error distributions of the final model, ECR-MobileNet-P (orange), and the baseline model (blue) for length and weight prediction.

**Table 1 animals-15-02443-t001:** Dataset partitioning and statistics.

Dataset	No. of Images	Total Length Range (cm)	Weight Range (g)	Mean Total Length ± SD (cm)	Mean Weight ± SD (g)
Training Set	1026	17.5–34.0	158–920	24.6 ± 2.6	419 ± 121
Validation Set	127	17.5–34.0	158–920	24.9 ± 3.1	427 ± 129
Test Set	131	17.5–34.0	158–920	24.4 ± 2.9	411 ± 118
Total	1284	17.5–34.0	158–920	24.6 ± 2.6	419 ± 121

**Table 2 animals-15-02443-t002:** Image data augmentation and preprocessing pipeline and parameters.

Data Subset	Resize	Horizontal Flip (p)	Random Rotation	Color Jitter (Brightness, Contrast, Saturation, Hue)	Random Grayscale (p)	Normalized
Training Set View 1	224 × 224	0.5	±20°	(0.3, 0.3, 0.3, 0.1)	0.1	√
Training Set View 2	224 × 224	0.5	±15°	(0.2, 0.2, 0.2, 0.0)	×	√
Validation/Test Set	224 × 224	×	×	×	×	√

The symbol ‘√’ denotes that the corresponding augmentation or preprocessing step was applied to the data subset. The symbol ‘×’ denotes that the step was not applied.

**Table 3 animals-15-02443-t003:** Baseline hyperparameter settings.

Parameter Type	Value
Optimizer	AdamW
Initial Learning Rate	1 × 10^−3^
Batch Size	16
Training Epochs	200
Learning Rate Scheduler	Cosine Annealing
Loss Function	Mean Squared Error (MSE)

**Table 4 animals-15-02443-t004:** Comparative results of the length prediction experiment.

Models	RMSE	MAE	MAPE	R^2^
Edgevit_xxs	0.6024	0.3245	1.39%	0.9576
Efficientformerv2_s2	0.4933	0.3162	1.32%	0.9716
Efficientnetv2	0.4979	0.2604	1.12%	0.9710
Ghostv3	0.5189	0.2935	1.25%	0.9685
Mobilenetv2	0.6178	0.3190	1.38%	0.9554
Mobilenetv3_large	0.4328	0.2571	1.07%	0.9781
Mobilenetv3_small	0.5234	0.3105	1.32%	0.9680
Mobilevit_s	0.6243	0.4255	1.75%	0.9545
Repvit	0.5502	0.3254	1.32%	0.9646
Resnet50	0.5465	0.2711	1.16%	0.9651
Shufflenetv2_x1.0	0.4457	0.2911	1.22%	0.9768
Tiny_vit	0.6109	0.3801	1.58%	0.9564
Vit_small	1.0401	0.7650	3.10%	0.8736
Fastvit_sa12	0.6380	0.4002	1.66%	0.9524
ECR-Mobilenet (ours)	0.4565	**0.2149**	**0.93%**	0.9757
ECR-Mobilenet-P (ours)	**0.4296**	0.2310	0.99%	**0.9784**

RMSE: Root mean square error; MAE: mean absolute error; MAPE: mean absolute percentage error; R^2^: coefficient of determination. For RMSE, MAE, and MAPE, lower values indicate better performance. For R^2^, higher values (closer to 1) indicate a better model fit.

**Table 5 animals-15-02443-t005:** Comparative results of the weight prediction experiment.

Models	RMSE	MAE	MAPE	R^2^
Edgevit_xxs	0.0274	0.0163	4.92%	0.9525
Efficientformerv2_s2	0.0239	0.0154	4.43%	0.9637
Efficientnetv2	0.0287	0.0139	4.44%	0.9479
Ghostv3	0.0228	0.0133	4.00%	0.9672
Mobilenetv2	0.0244	0.0134	3.80%	0.9621
Mobilenetv3_large	0.0234	0.0134	3.92%	0.9654
Mobilenetv3_small	0.0253	0.0153	4.61%	0.9595
Mobilevit_s	0.0269	0.0184	4.92%	0.9540
Repvit	0.0265	0.0158	4.46%	0.9554
Resnet50	0.0275	0.0134	4.20%	0.9522
Shufflenetv2_x1.0	0.0224	0.0140	4.05%	0.9680
Tiny_vit	0.0237	0.0156	4.47%	0.9643
Vit_small	0.0446	0.0308	7.86%	0.8741
Fastvit_sa12	0.0287	0.0187	5.33%	0.9479
ECR-Mobilenet (ours)	0.0228	0.0109	3.50%	0.9671
ECR-Mobilenet-P (ours)	**0.0202**	**0.0108**	**3.31%**	**0.9740**

RMSE: Root mean square error; MAE: mean absolute error; MAPE: mean absolute percentage error; R^2^: coefficient of determination. For RMSE, MAE, and MAPE, lower values indicate better performance. For R^2^, higher values (closer to 1) indicate a better model fit.

**Table 6 animals-15-02443-t006:** Comparative analysis of model complexity and lightweighting results.

Models	Parameters (M)	GFLOPs	CPU Latency (ms)	Memory (MB)
Edgevit_xxs	3.77	1.10	31.13	14.42
Efficientformer_s2	12.13	2.48	73.19	46.97
Efficientnetv2	20.18	5.44	74.45	77.56
Ghostv3	6.85	0.87	112.19	26.58
Mobilenetv2	2.22	0.61	29.29	8.62
Mobilenetv3_large	4.20	0.44	22.89	16.13
Mobilenetv3_small	1.51	0.11	12.30	5.84
Mobilevit_s	4.93	2.48	63.72	18.88
Repvit_0.9	6.40	2.27	53.29	24.65
Resnet50	23.51	8.26	66.14	89.89
Mhufflenetv2_x1.0	1.25	0.30	20.87	4.85
Tiny_vit	5.07	2.37	45.01	21.21
Vit_small	21.66	6.44	43.27	82.65
Fastvit_sa12	10.55	3.00	58.50	40.41
ECR-Mobilenet (ours)	0.93	0.12	14.46	3.59
ECR-Mobilenet-P (ours)	**0.52**	**0.07**	**10.19**	**2.00**

M: Million (10^6^); GFLOPs: Giga floating-point operations per second, a measure of computational complexity; ms: milliseconds; MB: megabytes. For all metrics in this table, lower values indicate higher efficiency.

**Table 7 animals-15-02443-t007:** Ablation study results for the length prediction task.

L1 Loss	ECA + Linear	AMCR	RMSE	MAE	MAPE	R^2^
×	×	×	0.5234	0.3105	1.32%	0.9680
√	×	×	0.5814	0.2771	1.21%	0.9605
×	√	×	0.5135	0.2969	1.26%	0.9692
×	×	√	0.5265	0.2952	1.25%	0.9676
√	√	×	0.4951	0.2397	1.04%	0.9714
√	×	√	0.5414	0.2552	1.10%	0.9658
×	√	√	0.5050	0.2651	1.13%	0.9702
√	√	√	**0.4565**	**0.2149**	**0.93%**	**0.9757**

RMSE: Root mean square error; MAE: mean absolute error; MAPE: mean absolute percentage error; R^2^: coefficient of determination. For RMSE, MAE, and MAPE, lower values indicate better performance. For R^2^, higher values (closer to 1) indicate a better model fit. ‘√’ indicates that the component was included in the model configuration; ‘×’ indicates it was excluded.

**Table 8 animals-15-02443-t008:** Ablation study results for the weight prediction task.

L1 Loss	ECA + Linear	AMCR	RMSE	MAE	MAPE	R^2^
×	×	×	0.0253	0.0153	4.61%	0.9595
√	×	×	0.0248	0.0123	3.88%	0.9609
×	√	×	0.0240	0.0141	4.22%	0.9634
×	×	√	0.0251	0.0145	4.36%	0.9600
√	√	×	0.0233	0.0116	3.55%	0.9665
√	×	√	0.0265	0.0119	3.85%	0.9554
×	√	√	0.0232	0.0134	4.04%	0.9657
√	√	√	**0.0228**	**0.0109**	**3.50%**	**0.9671**

RMSE: Root mean square error; MAE: mean absolute error; MAPE: mean absolute percentage error; R^2^: coefficient of determination. For RMSE, MAE, and MAPE, lower values indicate better performance. For R^2^, higher values (closer to 1) indicate a better model fit. ‘√’ indicates that the component was included in the model configuration; ‘×’ indicates it was excluded.

## Data Availability

Data are available upon request due to privacy considerations.

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
