# Peer review of "ECR-MobileNet: An Imbalanced Largemouth Bass Parameter Prediction Model with Adaptive Contrastive Regression and Dependency-Graph Pruning"

_animals, 2025, doi:10.3390/ani15162443_

Round 1
Reviewer 1 Report
Comments and Suggestions for Authors
This manuscript, “ECR-MobileNet:An Imbalanced Largemouth Bass ParameterPrediction Model with Adaptive Contrastive Regression andDependency-Graph Pruning” focuses on how AI technology can be used to accurately track the growth of largemouth bass for sustainable and profitable fish farming. This is a very interesting study that is important for achieving intelligent and modernized aquaculture. However, this research does not have the prospect of practical application, and the overall article is more suited to the direction of Artificial Intelligence, so it will be rejected with the following issues:
- the numerical accuracy of body length and weight prediction in the keywords in question should be added to the abstract to make it consistent with a paper in the main direction of a journal.
- Line 54-56 The scientific name of largemouth bass needs to be italicized and the most recent data on largemouth bass culture production should be provided.
- Line 163 What is a typical aquaculture environment? The aquaculture environment needs to be specified. The specific aquaculture background of the experimental fish used, e.g., body weight and length data, should be provided.
- Line 167-192 The following three experimental scenarios lack the introduction of the construction of specific scenes, when collecting image data, whether to shoot one by one individually or collectively need to be introduced. How to ensure that the fish are photographed to avoid the problem of inaccurate data collection caused by stress? Finally, can the sample size of 120 fish support the data collection, and why are there only 30 fish in experimental programs 2 and 3?
- Only scenario 2 of the above three scenarios can represent the aquaculture environment in the actual marketing process, while the other scenarios do not have the characteristics of simulating the actual aquaculture environment.
- Line 204 How to measure the weight and length of live fish immediately, the numerical measurement is unstable under the state of fish stress, the data collection is not accurate, the weight should be accurate to milligrams.
- Line 208 How to correlate the measurement data with the image accurately?
- Line 217-224 How to obtain experimental samples with such a large coefficient of variation in body weight? In the retail sector, largemouth bass with only 158g will not be sold, and in the actual culture process, there are very few cultured individuals of 158-920g, and the authors need to provide the specific growth data of the 120 samples.
Reviewer 2 Report
Comments and Suggestions for Authors
This study, 'ECR-MobileNet: An Imbalanced Largemouth Bass Parameter Prediction Model,' presents outstanding research on precise, non-destructive fish monitoring. It addresses dual challenges: traditional methods cause stress, while computer vision struggles with prediction biases and accuracy-lightweighting balance. The manuscript is well-structured, with clear data presentation and insightful interpretation, providing valuable findings.
However, to further enhance the manuscript's quality, I have identified several issues (mentioned in pdf) that necessitate major revisions.

Reviewer 3 Report
Comments and Suggestions for Authors
Comments and Suggestions
- The paper presents a deep learning framework for fish biometric monitoring - clearly targeting precision aquaculture applications.
- Clear Problem Statement: You effectively articulate the dual challenges (stress/loss from contact methods, bias/deployment bottlenecks in CV) and the significance of precise, non-destructive monitoring.
- Convincing Solution: The proposed framework (ECR-MobileNet) directly addresses the stated challenges:
- Lightweight Base: MobileNetV3-Small is an excellent choice for edge deployment.
- Addressing Imbalance: The novel AMCR loss function tackling imbalance in regression is a significant contribution beyond standard classification approaches.
- Mitigating Bias: Explicitly linking AMCR to reducing prediction bias caused by imbalance is crucial.
- Optimizing Deployment: DepGraph pruning tackles the accuracy-lightweighting trade-off head-on.
- Strong Results: The performance metrics (R², RMSE) are exceptionally high for both length and weight prediction, convincingly demonstrating the effectiveness of the proposed framework.
- Impressive Efficiency: The parameter count (0.52M), computational load (0.07 GFLOPs), and CPU latency (10.19 ms) are outstanding and strongly support the claim of edge deployability and Pareto optimality.
- Comprehensive Benchmarking: Outperforming 14 mainstream benchmarks provides strong comparative evidence.
- Broader Impact: Clearly stating the contributions to both aquaculture (edge-deployable solution) and methodology (imbalanced regression, task-oriented compression) elevates the significance.
- Quantify "Imbalanced Data":
- Current: Mentioned as a cause of bias but not quantified.
- Suggestion: Briefly characterize the imbalance in your dataset (e.g., "characterized by a long-tailed distribution of fish sizes" or "where small/large fish were underrepresented"). This strengthens the justification for AMCR.
- Clarify "Multi-scene":
- Current: Describes the dataset but leaves "multi-scene" ambiguous.
- Suggestion: Briefly specify what constitutes a "scene" (e.g., different tank backgrounds, lighting conditions, water turbidity levels, camera angles?). This clarifies the robustness claim.
- Elaborate Slightly on ECA's Role:
- Current: States it enhances feature discriminability.
- Suggestion: Briefly connect it to the specific challenge (e.g., "...crucial for distinguishing fish features under variable water conditions" or "...improves focus on salient morphological features amidst clutter").
- Highlight "Multi-dimensional" in AMCR:
- Current: Mentions "multi-dimensional regression".
- Suggestion: Explicitly state that this refers to predicting both length and weight simultaneously within the loss function, if that's the case. This clarifies the innovation beyond single-target regression.
- Strengthen DepGraph Justification:
- Current: States it's for synergistic optimization.
- Suggestion: Briefly hint why dependency-graph-based pruning is particularly suitable for this task over other pruning methods (e.g., preserves task-critical feature interactions identified during training).
- Contextualize Performance Numbers (Slightly):
- Current: Presents raw R²/RMSE for ECR-MobileNet-P.
- Suggestion: While the numbers are excellent, adding a single sentence placing them in context could help readers unfamiliar with fish biometrics benchmarks (e.g., "These results represent a significant improvement over the state-of-the-art for underwater fish biometrics," or compare the RMSE to a typical acceptable error margin in aquaculture practice if known).
- Edge Deployment Evidence:
- Current: Claims "edge-deployable" based on model size/latency.
- Suggestion: Briefly mention the platform used for the CPU latency test (e.g., "on a common edge device like a Raspberry Pi 4" or "on an Intel i5-XXXX CPU"). This grounds the claim in reality.
- Broader Applicability Hint:
- Current: Focused on largemouth bass.
- Suggestion: Add a phrase suggesting the framework's potential applicability beyond this species (e.g., "...providing a solution for largemouth bass, with methodology applicable to other aquaculture species").
- Compare with others. In this aspect, I would like to advise the authors to refer to the following papers in their revised version.
- Thi Thi Zin, T. Morimoto, Naraid Suanyuk, T. Itami and Chutima Tantikitti, "Image technology based detection of infected shrimp in adverse environments", Songklanakarin Journal of Science and Technology, 44 (1), pp.112-118, DOI:14456/sjst-psu.2022.17, Jan. 2022.
- Morimoto, Thi Thi Zin, T. Itami, "A Study on Abnormal Behavior Detection of Infected Shrimp", Proc. on 2018 IEEE 7th Global Conference on Consumer Electronics, GCCE 2018, 2018, pp. 818–819, 8574860.
Comments on the Quality of English Language
The paper is well written.
Round 2
Reviewer 1 Report
Comments and Suggestions for Authors
In the revised manuscript the authors have responded well to the review comments, but there is still room for further improvement, as described in the following revisions:
- Please re-correct the latest production of largemouth bass, the data provided by the author is not the latest official statistics.
- The authors have provided some data on the culture environment, but it is not the typical culture environment data, which needs to be re-corrected. Table 1 only provides statistical data, please provide the precise growth data of 120 experimental individuals in the supplementary table.
- Add the specific difference between scenarios 1 and 2, and how to design these two scenarios, and add the concentration of MS-222.
- How can Program 1 and 3 achieve the monitoring application in the farm, the depth position of the camera fixed underwater needs to be supplemented, in addition to the actual aquaculture process can not be realized in the observation of Program 3, please rationally explain the design of this program.
- Please provide the reference basis for the accuracy level of weight measurement to meet the standard requirements of aquaculture management, and whether the weight accuracy of 0.1g is achievable.
- Please provide labeled photographs in the accompanying figures.
- The authors collected commercial fish from juvenile to market size, but what is the proportion of samples collected, providing specific measurements of 120 fish can explain this well.
In addition, the authors should pay more attention to the formatting of the text when responding to comments, so please double-check and change it. For example, the formatting of “ Comments 6”.
Reviewer 2 Report
Comments and Suggestions for Authors
Accept
Author Response
Thank you for your positive feedback.